# Chemotherapeutic Regimens and Chemotherapy-Free Intervals Influence the Survival of Patients with Recurrent Advanced Epithelial Ovarian Carcinoma: A Retrospective Population-Based Study

**DOI:** 10.3390/ijerph18126629

**Published:** 2021-06-20

**Authors:** Hsin-Ying Huang, Chun-Ju Chiang, Yun-Yuan Chen, San-Lin You, Heng-Cheng Hsu, Chao-Hsiun Tang, Wen-Fang Cheng

**Affiliations:** 1School of Health Care Administration, Taipei Medical University, Taipei 106, Taiwan; g559103001@tmu.edu.tw (H.-Y.H.); chtang@tmu.edu.tw (C.-H.T.); 2Graduate Institute of Epidemiology and Preventive Medicine, College of Public Health, National Taiwan University, Taipei 100, Taiwan; ruru.chiang@cph.ntu.edu.tw; 3Taiwan Cancer Registry, Taipei 100, Taiwan; 4Taiwan Blood Services Foundation, Taipei 100, Taiwan; yunyuan@blood.org.tw; 5School of Medicine, College of Medicine and Big Data Research Centre, Fu-Jen Catholic University, New Taipei City 242, Taiwan; yousanlin@gmail.com; 6National Taiwan University Hospital Hsin-Chu Branch, Hsin-Chu City 300, Taiwan; 7Graduate Institute of Clinical Medicine, National Taiwan University, Taipei 100, Taiwan; 8Department of Obstetrics and Gynecology, National Taiwan University, Taipei 100, Taiwan; 9Graduate Institute of Oncology, College of Medicine, National Taiwan University, Taipei 100, Taiwan

**Keywords:** ovarian carcinoma, recurrence, paclitaxel, liposomal doxorubicin, topotecan, chemotherapy-free interval

## Abstract

We aimed to evaluate factors influencing the outcomes of patients with platinum-sensitive recurrent epithelial ovarian carcinoma (EOC). Patients with advanced-stage EOC, who received debulking surgery and adjuvant chemotherapy for recurrence, were obtained from the National Health Insurance Research database of Taiwan between 2000 and 2013. A total of 1038 patients with recurrent advanced-stage EOC were recruited. The platinum + paclitaxel (PT) group had the best five-year overall survival (OS) compared with the other three groups (*p* < 0.001). The hazard ratios (HRs) of five-year OS for the platinum + liposomal doxorubicin (PD), topotecan (TOP), and pegylated liposomal doxorubicin (PLD) groups were 1.21 (*p* = 0.07), 1.35 (*p* = 0.016), and 1.80 (*p* < 0.001), respectively, compared with the PT group. The PT group also had lower hazard ratios of five-year OS for patients with platinum therapy-free interval (TFIp) between 6 and 12 months compared with the other three groups (*p* < 0.0001). However, the HRs of five-year OS did not differ between the PT and PD groups in patients with TFIp >12 months. Patients with TFIp >12 months had lower HRs of five-year OS compared with those with TFIp of 6–12 months, regardless of whether they were treated with platinum-based (*p* = 0.001) or non-platinum-based (*p* = 0.003) regimens. Chemotherapeutic regimens and TFIp influenced the outcomes of patients with recurrent EOC. For patients with TFIp of 6–12 months, the PT regimen is the first choice based on their best overall survival result. For patients with TFIp >12 months, either platinum-based or non-platinum regimens could be used because of their similar excellent overall survival.

## 1. Introduction

According to global research and statistics, epithelial ovarian carcinoma (EOC) is the seventh most common cancer in women and the eighth most common cause of cancer-related deaths, with five-year survival rates below 45% worldwide [1]. The global age-standardized rate (ASR) of ovarian cancer is highest in Central/Eastern Europe (11.9/100,000), followed by Northern Europe (9.2/100,000), North America (8.4/100,000), Western Europe (7.0/100,000), Australia (6.9/100,000), and Middle Africa (3.8/100,000) [2]. The ASR of ovarian cancer is 9.18/100,000 in Taiwan [3], which is higher than that of East Asia (5.8/100,000) [2]. Ovarian cancer is also the seventh most common invasive cancer in women, and the eighth leading cause of cancer-related deaths in women in Taiwan [4]. In the past two decades, the incidence of EOC has increased and the age of diagnosis has decreased in Taiwan. The incidence in Taiwan is comparable to those of Western countries [5].

Patients with ovarian cancer are usually diagnosed with advanced disease, because early diagnosis is difficult due to the lack of obvious initial symptoms. For patients with low residual disease (all lesions <1 cm in size after debulking surgery), the risk of recurrence is 60%–70%. However, after surgery, the risk of recurrence is about 80%–85% in women with large-volume residual disease [6]. Despite initial debulking surgery followed by platinum-based adjuvant, 50%–75% of EOC patients with advanced disease relapse and need salvage chemotherapy. When EOC recurs, the platinum therapy-free interval (TFIp, period between the last front-line platinum-based chemotherapy and first salvage chemotherapy re-introduction) is an important factor for choosing the optimal regimen following salvage chemotherapy. Patients with recurrent EOC are considered to have platinum-sensitive (PS) or platinum-resistant (PR) disease based on the length of the TFIp. The Fifth Ovarian Cancer Consensus suggested that the specific time from last platinum should be reported to avoid arbitrary division of platinum-sensitive (PS) or platinum-resistant (PR) patients [7]. However, for research purposes, the current clinical trials still utilize the classic classification [8,9,10]. If the TFIp is >6 months, the patients are deemed PS; within this group, patients with a TFIp between 6 and 12 months are considered partially PS and patients with a TFIp longer than 12 months are considered complete PS. If the TFIp is ≤6 months, patients are deemed PR. Thus, the TFIp, namely, the time between the last complete cycle of platinum therapy and evidence of disease recurrence, is an important, significant, and reliable predictor of response to second-line chemotherapy [6].

Platinum single-agent or platinum-based combination chemotherapies are recommended as treatment for patients with PS EOC [11,12]. A longer TFIp is associated with better outcomes. Specifically, the response rates for TFIp of 5–12 months, 13–24 months, and >24 months are 27%, 33%, and 59%, respectively [6]. Therefore, it was hypothesized that extending the platinum-free interval (PFI) with a non-platinum-based regimen could potentially improve survival outcomes [13]. Platinum alone, platinum combined with paclitaxel or other cytotoxic drugs including liposomal doxorubicin or gemcitabine, or non-platinum chemotherapy regimens such as topotecan or liposomal doxorubicin alone are options for PPS (partial platinum sensitive) patients with TFIp between 6 and 12 months [12,14,15,16,17,18,19,20]. Few studies have focused on comparing treatment outcomes of patients with PPS and patients with PS recurrent EOC. Pujade-Lauraine et al. reported that the combination of carboplatin and liposomal doxorubicin was superior than carboplatin and paclitaxel in patients with PS recurrent EOC in terms of progression-free survival (PFS) [17]. These prior studies were well-designed prospective studies or retrospectively chart-reviewed studies with small sample sizes; however, the chemotherapy response and outcomes in these patients with PS recurrent EOC, especially those with PPS, have not yet been fully elucidated.

Therefore, we conducted a population-based analysis to determine if patients with PPS and PS recurrent advanced EOC treated with platinum-based regimens had better responses and outcomes than those treated with non-platinum chemotherapy regimens, and if different platinum-based combination chemotherapy regimens led to different responses and outcomes.

## 2. Patients and Methods

### 2.1. Study Design

A retrospective cohort of women with platinum-sensitive recurrent ovarian cancer was established through computerized linkages of the National Health Insurance Research Database (NHIRD) program with several national databases containing information on demographic characteristics and diagnosis, inpatient and outpatient medical order files, the catastrophic illness registry, and the death certificate registry. The NHIRD program is a social insurance program organized by the government under the jurisdiction of the Ministry of Health and Welfare, which provides co-payment reimbursement to patients. The patients in this study included advanced-stage EOC patients who had disease progression more than 6 months after previous platinum-based chemotherapy. In this retrospective cohort study, overall survival (OS) was compared in PS patients with recurrent advanced EOC who were administered the following four major chemotherapy regimens, as recommended by clinical guidelines in Taiwan [21], including platinum analogues combined with paclitaxel, platinum analogues combined with liposomal doxorubicin, topotecan alone, and liposomal doxorubicin alone for the first chemotherapy treatment after cancer recurrence. The International Federation of Gynecology and Obstetrics classifications were used for the staging of ovarian cancer [22]. Stage III and IV were defined as advanced-stage disease.

### 2.2. Study Population

The patients in this study included advanced-stage EOC patients with disease progression more than 6 months after previous platinum-based chemotherapy. The patients who fulfilled the criteria were further identified from the catastrophic illness registry. Ovarian cancer patients who received gynecological surgery (staging or debulking surgery) between 1 January 2000 and 31 December 2011 were included in this study. To identify patients with advanced-stage ovarian cancer (stage III or IV), those who underwent frontline debulking surgery (standard surgery for stage III or IV) with adjuvant chemotherapy were recruited. The National Health Insurance (NHI) in Taiwan only reimburses platinum and paclitaxel for the frontline adjuvant chemotherapy of stage III or stage IV ovarian cancer. Thus those who received platinum-plus-paclitaxel-containing regimens reimbursed by the NHI followed by debulking surgery were cases of advanced-stage and thus included. The debulking operation included abdominal total hysterectomy, bilateral salpingo-oophorectomy, omentectomy, and retroperitoneal lymphadenectomy. The inclusion criteria were patients who received surgery for ovarian cancer during 2000–2011 and had no history of any other cancer type. The exclusion criteria were (1) first cycle of chemotherapy >60 days after surgery, (2) interval of each chemotherapy cycle >60 days, (3) less than six cycles of chemotherapy (considered incomplete treatment), (4) more than nine chemotherapy cycles (considered persistent disease), (5) single-agent used as first-line adjuvant chemotherapy, (6) adjuvant chemotherapy regimens other than platinum and paclitaxel, (7) interval between each cycle of adjuvant chemotherapy >60 days, and (8) interval between the last date of the last cycle of adjuvant chemotherapy and the first cycle of salvage chemotherapy <6 months (considered platinum resistance). The eligible patients in this study received six to nine cycles of platinum + paclitaxel chemotherapy within 60 days after surgery as adjuvant chemotherapy, and had >6 months of clinical remission before receiving salvage chemotherapy after disease relapse.

### 2.3. Second-Line Salvage Chemotherapy

Patients with PS recurrent ovarian cancer who discontinued treatment after the last cycle of adjuvant chemotherapy longer than 6 months ago were treated with single or combination chemotherapy regimens, and were defined as recurrent PS EOC patients. The chemotherapy regimens after recurrence were recorded and regarded as second-line chemotherapy or first-line salvage chemotherapy after disease relapse. Based on the major chemotherapy agents, patients with PS advanced-stage ovarian cancer were grouped into the following four treatment regimens according to their first chemotherapy for second-line treatment: PT, platinum + liposomal doxorubicin (PD), topotecan (TOP) alone, and pegylated liposomal doxorubicin (PLD) alone. The study flow chart is shown in Figure 1. The patients were further stratified into CPS (TFIp > 12 months) and PPS (TFIp between 6 and 12 months) groups.

### 2.4. Statistical Analyses

The frequencies of characteristics in the PT, PD, TOP, and PLD groups, including age, TFIp, Charlson comorbidity index, hospital level, and residential areas, were retrieved from the NHI research database and compared using the Chi-squared test to identify possible confounding factors. The Charlson comorbidity index is a method for predicting mortality by classifying or weighing comorbid conditions such as myocardial infarction, congestive heart failure, peripheral vascular disease, cerebrovascular disease, dementia, chronic lung disease, connective tissue disease, ulcer disease, mild liver disease, diabetes, diabetes associated with multiple organ failure, hemiplegia, moderate or severe kidney disease, leukemia, lymphoma, moderate or severe liver disease, and acquired immunodeficiency syndrome. If the main diagnosis or secondary diagnosis was one of the aforementioned diseases (excluding cancer), we included a calculation of comorbidity. The Charlson comorbidity index was classified by scores greater than 1 or equal to 0 in this study. The number of cycles of first-line chemotherapy was calculated based on the orders registered in the inpatient and outpatient medical order files. If a new course of chemotherapy was started at least 1 month after the completion of adjuvant chemotherapy, and cytotoxic drugs such as cisplatin, carboplatin, paclitaxel, topotecan, and liposomal doxorubicin reimbursed by the NHI were administered, recurrence was considered to have occurred. In this study, the TFIp was >6 months. The main outcomes evaluated were death and cancer recurrence. The follow-up period for TFIp in each patient was calculated from the end of the chemotherapy course to the date a new course (chemotherapy for cancer recurrence) was initiated. Death ascertainment was retrieved from the death certificate. The period for overall survival (OS) was from the date of the first cycle of salvage chemotherapy after cancer relapse to the date of death or alive until 31 December, 2013, whichever occurred first. The OSs of different groups were estimated using the Kaplan–Meier method, and the differences in survival curves among the four groups were tested by the log-rank test. To compare the effectiveness of PT, PD, TOP, and PLD, hazard ratios (HRs) of different regimens adjusted by confounding factors and 95% confidence intervals (CIs) were estimated using Cox proportional hazards models. All statistical analyses were performed using SAS 9.3 (SAS Institute, Cary, NC, USA).

## 3. Results

### 3.1. Patient Characteristics

A total of 1038 patients with recurrent advanced EOC, including 605 who received platinum analogues combined with a paclitaxel regimen (PT group), 204 who received platinum analogues combined with and pegylated liposomal doxorubicin (PD group),109 who received TOP alone, and 120 who received pegylated liposomal doxorubicin (PLD) alone, were included in this study. The basic characteristics of the 1038 patients are shown in Table 1. Of the patients, 63.58% (660/1038) were older than 50 years of age. The PT group had a higher percentage (65.8%) of patients with a TFIp >12 months compared with the other groups (*p* < 0.0001, chi-square test).

### 3.2. Patients Treated with Platinum Combined with Paclitaxel Had the Best Five-Year OS

As shown in Figure 2, the five-year OS of the 1038 patients with recurrent advanced-stage EOC, who received different chemotherapy regimens, were further analyzed. The estimated probability of five-year OS was 31% in the PT group, 18% in the PD group, 16% in the TOP group, and 10% in the PLD group. The PT group also showed better five-year OS compared with the PD group (*p* < 0.001, by log-rank test). The HRs of five-year OS after adjusting for other confounding factors including age, progression-free interval, hospital level, and residential areas were further evaluated. Compared with the PT group, the HRs of five-year OS for the PD, TOP, and PLD groups were 1.21 (95% CI: 1.02–1.61, *p* = 0.07), 1.35 (95% CI: 1.06–1.73, *p* = 0.016), and 1.80 (95% CI: 1.43–2.27, *p* < 0.001), respectively. PD had better OS than that of PLD (*p* < 0.0001) or TOP alone (*p* = 0.014) in terms of two- year OS (Figure 2). Our results indicated that PT regimens for recurrence led to better survival than obtained with treatment with TOP or PLD alone. The median survivals of PT, PD, TOP, and PLD were 33.6, 28.2, 21.8, and 17.0 months in this study (Table 2).

### 3.3. TFIp Impacted Five-Year OS in Patients with PS Recurrent Advanced EOC

We further analyzed the impact of TFIp on the outcomes of patients with recurrent advanced EOC treated with different chemotherapy regimens after recurrence. The patients were divided into two groups according to the length of the TFIp interval (6–12 months and >12 months) for the following analysis. As shown in Table 3, when using the PT group as a reference, the HRs of five-year OS of patients with TFIp between 6 and 12 months were significantly lower than those in the PD (HR: 1.36, 95% CI: 1.03–1.79, *p* = 0.031), TOP (HR: 1.41, 95% CI: 1.06–1.89, *p* = 0.018), and PLD (HR: 1.86, 95% CI: 1.42–2.43, *p* < 0.0001) groups. On the other hand, there were no differences in HRs of five-year OS among patients with TFIp >12 months in the groups (PD group—HR: 1.13, 95% CI: 0.82–1.57, *p* = 0.46; TOP group—HR: 1.36, 95% CI: 0.80–2.28, *p* = 0.25; PLD group—HR: 1.59, 95% CI: 0.96–2.617, *p* = 0.07) (Table 3). These results indicate that among patients whose TFIp ranged from 6 to 12 months, those treated with PT had the best five-year OS compared with the other three regimens. However, patients with TFIp >12 months had similar five-year OS when treated with the respective chemotherapy regimens.

### 3.4. Patients with TFIp > 12 Months Had Better Five-Year OS Than Those with TFIp of 6–12 Months 

Next, we evaluated if the chemotherapy regimens influenced the outcomes of patients with different durations of TFIp. As shown in Table 4, patients with TFIp >12 months had lower HRs of five-year OS than those with TFIp of 6–12 months (HR: 1.00) treated with platinum (HR: 0.57, 95% CI: 0.4–0.68, *p* < 0.001) or non-platinum (HR: 0.56, 95% CI: 0.38–0.82, *p* = 0.003) chemotherapy regimen(s). Our results indicate that TFIp is an important factor for predicting the chemotherapy response and outcome of patients with recurrent ovarian cancer, when re-introducing chemotherapy.

## 4. Discussion

Our results showed that patients with recurrent epithelial ovarian cancer treated with platinum-based combination chemotherapy had significantly better overall survival than those treated with a non-platinum single-agent, regardless of whether it was topotecan or liposomal doxorubicin. In addition, platinum with paclitaxel had significantly better OS than platinum with liposomal doxorubicin did in patients with recurrent disease and TFIp between 6 and 12 months. Whereas, the regimens of both platinum with paclitaxel and platinum with liposomal doxorubicin had similar OS in patients with TFIp >12 months.

It is unknown if extending the PFI with non-platinum agents can really improve the response after re-introducing platinum-based chemotherapy and then result in survival benefits in these patients. Patients with longer platinum-free interval have a better outcome in recurrent ovarian cancer [23]. The MITO8 phase III study tested if prolonging platinum-free interval in patients with ovarian cancer recurring between 6 and 12 months after previous platinum-based chemotherapy could improve the survival. Similar median overall survivals were noted between non-platinum and platinum-based chemotherapeutic groups (21.8 versus 24.5 months, *p* = 0.06) [24]. Whereas, our results showed that patients with a TFI between 6 and 12 months treated with non-platinum regimens had poorer outcomes than those directly re-treated with platinum-based regimens in this Asian real-world population study (Figure 2). The different ethnicities, initial stages, histological types, and status of primary or secondary debulking surgeries could influence MITO-8 and our results. There was a poorer five-year OR when treated with TOP (HR: 1.41, 95% CI: 1.06–1.89, *p* = 0.018) or PLD (HR: 1.86, 95% CI: 1.42–2.43, *p <* 0.0001) alone compared with the PT group. In addition, the platinum-based doublet regimens (PT or PD) led to better five-year overall survival than non-platinum single regimens (TOP or PLD) did in all of the patients (Figure 2). Because these women had good response to frontline platinum-based chemotherapy, the re-introduction of platinum-based combination therapy could be expected to generate better response than achieved with the non-platinum regimen alone like our results.

In patients with recurrent ovarian cancer with TFIp of 6–12 months, those treated with PT had improved survival compared with those treated with PD. Our results indicated that the PD group had worse five-year OS than the PT group did (HR: 1.36, 95% CI: 1.03–1.79, *p =* 0.031, Table 3) in patients with TFIp of 6–12 months. The CALYPSO trial revealed similar OS outcomes between the PT and PD groups [16]; however, patients in the PD group experienced less toxicity and adverse events compared with those in the PT group [17]. Thus, platinum with liposomal doxorubicin instead of platinum with paclitaxel is recommended for patients with recurrent ovarian cancer with TFIp of 6–12 months, especially those with paclitaxel-related toxicities [17,22]. Our study had different results than the CALYPSO trial, for the following possible reasons. First, the CALYPSO trial analyzed all of the recruited patients, although 6% and 15% of patients in the PD and PT groups, respectively, did not complete six cycles of chemotherapy [17]. In contrast, in our study, all patients in the PD and PT groups received at least six cycles of chemotherapy. Patients who receive less than six cycles of chemotherapy may have a poorer performance status. Second, the PLD used in the CALYPSO trial differed from that used in our study, as the CALYPSO trial used Caelyx (Eli Lily, Bruxelles, Belgium) [17,25] and our study used Lipo-Dox (TTY BioPharm, Taipei City, Taiwan) [15,26]. Third, the CALYPSO trial was an international clinical trial in which the majority of participants were Caucasians, whereas our study was a daily practice, observational study that comprised a large majority of Mongolians. Several studies have shown disparity in treatment outcomes between different ethnicities in gynecology oncology, such as the dose-dense regimen in front-line ovarian cancer [27,28] and maintenance regimen of pazopanib in ovarian cancer [29]. Based on our results, we recommend PT as the first-choice regimen for patients with recurrent EOC with TFIp of 6–12 months. PD can be considered for these patients with intolerable paclitaxel-related toxicities [30].

Platinum-based combination chemotherapy with paclitaxel or liposomal doxorubicin had similar survival benefit for patients with recurrent ovarian cancer with TFIp >12 months. The HR of five-year survival was 1.13 (95% CI: 0.82–1.57, *p* = 0.46, Table 3) in the PD group as compared with PT group. Similar to our results, the CALYPSO trial also showed that the PT and PD regimens resulted in similar OS in patients with TFIp >12 months (HR: 0.99, 95% CI: 0.91–1.21, *p* = 0.90) [13,16]. However, in that trial, the PT groups had significantly higher incidences of grade 3–4 neutropenia, ≥grade 2 alopecia, ≥grade 2 sensory neuropathy, allergic reaction, and arthralgia/myalgia compared with the PD group [17]. Thus, we recommend PD rather than PT regimens as the first-choice chemotherapy regimen for patients with recurrent ovarian cancer with TFIp >12 months, because PD regimens had similar outcomes and less toxicities.

The paradigm of platinum-sensitive recurrent ovarian cancer treatment has changed dramatically in the last 10 years. Four randomized control trials with the usage of bevacizumab [9,31,32] and another three with the maintenance of PARP inhibitor [10] have been published, all with PFS benefits, with some having immature data on OS. Several studies have shown that PARPi accounted for a significant improvement of PFS in both recurrent and primary ovarian cancer patients, independently from patients’ BRCA mutational status [33].

What’s more, low- and middle-income countries (LMIC) may not have accessibility to these new agents [34], and those that have access to these agents might still have issues with reimbursement and costs, creating a barrier for patients to be treated by these new agents [35]. Thus, chemotherapy treatment is still an important pillar of the treatment in platinum-sensitive recurrent ovarian cancer.

This study had several limitations and strengths. The study was retrospective and observative in nature, and not a randomized clinical study. However, this weakness is also a strength of this study, as the patients reflected those who are treated in the real world. In addition, the data were obtained from the majority of the patients, in contrast to randomized clinical trials in which many patients are excluded due to the stringent exclusion criteria. Although it has been suggested that platinum sensitivity should not be classified based on a six-month cut-off from the last platinum-based chemotherapy, and other factors such as tumor biology, tumor histology, prior response, persistent toxicity, symptoms, and patients’ preferences should be taken into account together for the choice of chemotherapeutic regimens [7], currently several clinical trials still implement the concept of platinum-free interval, because it is a more feasible way to categorize patients for their research purposes [8,9,10]. The strength of our study is that, in contrast to multiple studies of platinum-sensitive recurrence ovarian cancer that did not report or had immature OS data (Table 2) [7,8,10,36,37,38], it provided the overall survival data of these patients. The majority (more than 99%) of the recruited patients in this study were Mongolian ethnicity, an ethnic group that has not been well represented in previous clinical trials. We recommend more additional prospective clinical studies or trials on Asian people to validate our results in our future.

For recurrent ovarian cancer patients residing in a country where new agents are not available or for whom surgery is not suitable, we propose a treatment regimen for patients with platinum-sensitive recurrent ovarian cancer (Figure 3). For patients with TFIp of 6–12 months, the PT regimen is the first choice based on its best overall survival result. For patients with TFIp >12 months, either platinum-based or non-platinum regimens could be used because of their similar excellent overall survival.

## 5. Conclusions

Chemotherapeutic regimens and TFIp influenced the outcomes of patients with recurrent EOC. For patients with TFIp of 6–12 months, the PT regimen is the first choice based on their best overall survival result. For patients with TFIp >12 months, either platinum-based or non-platinum regimens could be used because of their similar excellent overall survival.

## Figures and Tables

**Figure 1 ijerph-18-06629-f001:**
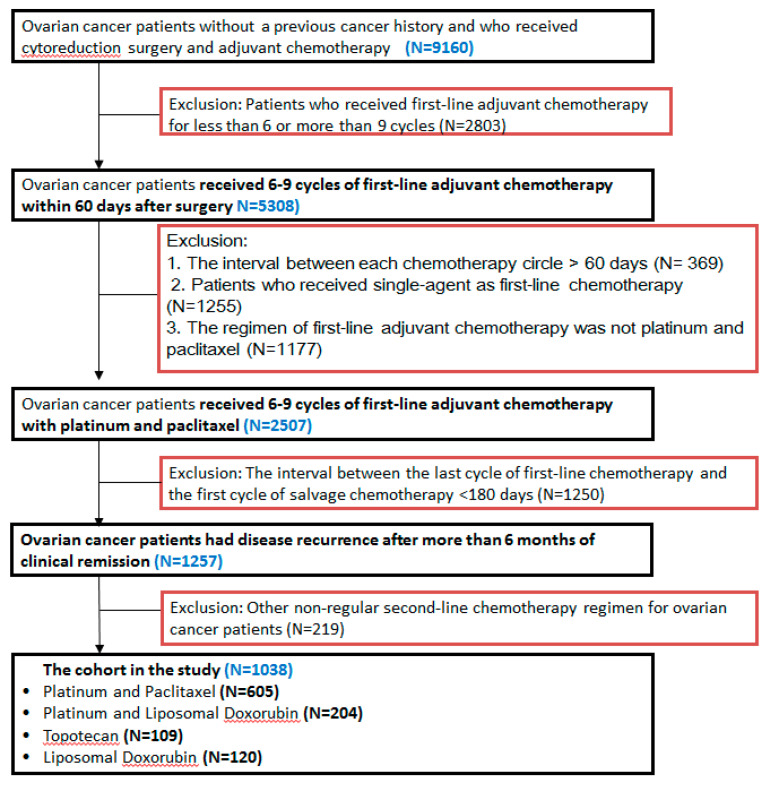
The flowchart of this study design.

**Figure 2 ijerph-18-06629-f002:**
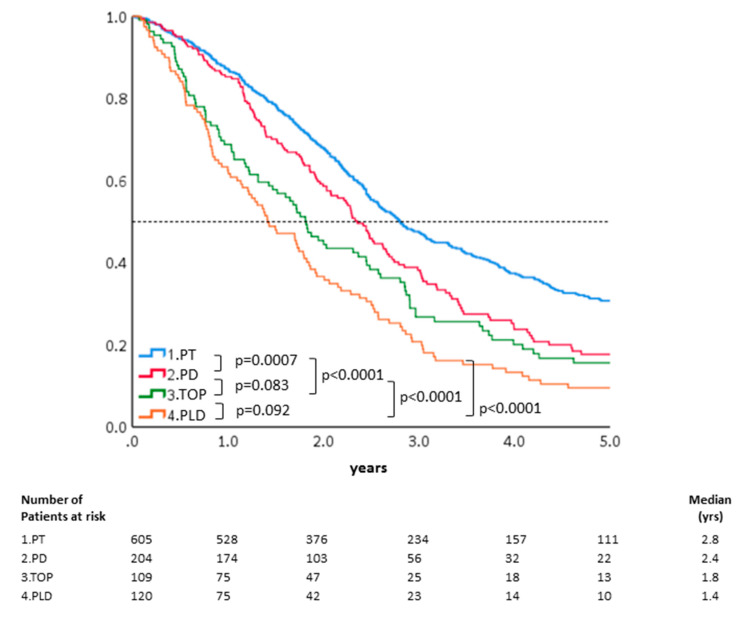
OS of the 1038 patients treated with different chemotherapeutic regimens. The estimated probability of five-year OS was 31% in the PT group, 18% in the PD group, 16% in the TOP group, and 10% in the PLD group. The PT group also showed better five-year OS compared with the PD group (*p* < 0.001, by log-rank test).

**Figure 3 ijerph-18-06629-f003:**
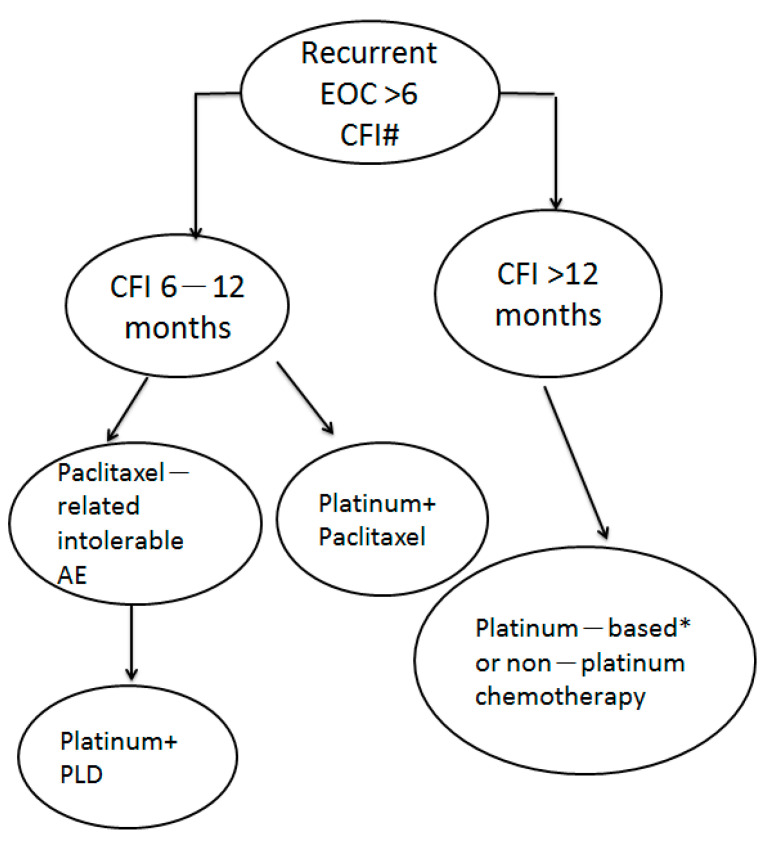
The proposed treatment of patients with PS recurrent ovarian cancer with different durations of TFIp. #: new agents are not available and patients not suitable for surgery. * platinum + liposomal doxorubicin could be used due to paclitaxel-induced intolerable adverse events such as peripheral neuropathy.

**Table 1 ijerph-18-06629-t001:** Characteristics of 1038 recurrent advanced-stage EOC patients.

Characteristics	PT (N = 605)		PD (N = 204)		TOP (N = 109)		PLD (N = 120)		*p*
	N	%	N	%	N	%	N	%	%
**Age**									
**Mean (SD)**	53.64 (9.94)		54.34 (10.68)		54.70 (11.42)		54.81 (10.81)		0.53
≤40	37	6.1	18	8.8	11	10.1	7	5.8	0.27
41–50	195	32.2	52	25.5	26	23.9	32	26.7	
51–60	220	36.4	77	37.8	36	33.0	43	35.8	
>60	153	25.3	57	27.9	36	33.0	38	31.7	
**TFIp (months)**									<0.001
6–12 (N = 504)	207	34.2	118	57.84	82	75.2	97	80.8	
>12 (N = 534)	398	65.8	86	42.16	27	24.8	23	19.2	
**Charlson Comorbidity** **Index**									0.52
Score = 0	545	90.1	184	90.2	96	88.1	103	85.8	
Score ≥1	60	9.9	20	9.8	13	11.9	17	14.2	
**Hospital Level**									<0.001
Public medical center	220	36.4	128	62.8	38	34.9	46	38.3	
Public non-medical center	11	1.8	1	0.5	0	0.0	2	1.7	
Private medical center	256	42.3	60	29.4	36	33.0	53	44.2	
Private non-medical center	118	19.5	15	7.3	35	32.1	19	15.8	

PT: platinum + paclitaxel; PD: platinum+ liposomal doxorubicin; TOP: topotecan; PLD: liposomal doxorubicin; SD, standard deviation; TFIp, platinum therapy-free interval.

**Table 2 ijerph-18-06629-t002:** Comparison of the different trials of PSROC and our current study.

	N	Previous Lines of Therapy		TFIp		PFS		OS (*p*-Value)	HR HR, P
**CALYPSO**		1	2	6–12	>12		0.82 (0.005)		0.99 (0.94)
PD	466	87.6%	12.4%	35%	65%	11.3		30.7	
PT	507	82.6%	17.3%	36.1%	63.9%	9.4		33	
**HECTOR**							n/a (0.414)		n/a (0.163)
TP	275	95.6%	20%	64%	36%	10		25	
GC/PT/PD	275	97.1%	18.9%	65.5%	33.5%	10		31	
**MITO-8**							1.41 (0.025)		1.38 (0.06)
PC	102	94.4%	5.6%	100%	0	16.4		24.5	
PLD	97	93.5%	6.5%	100%	0	12.8		21.8	
**OCEANS**							0.48 (<0.0001)		
PGbev	242	100%		41.3%	42.1%	12.4		n/a	
PG	242	100%		58.7%	57.9%	8.4		n/a	
**GOG213**				6–12	>12		0.628 (<0.001)		0.829 (0.056)
PCbev	337	100%		27%	73%	13.8		42.2	
PC	337	100%		25%	75%	10.4		37.3	
**AGO**							0.861 (0.012)		
PDbev		100%	0	31%	69%	13.3		n/a	
PGbev		100%	0	31%	69%	11.6		n/a	
**MITO16B**							0.51 (0.0001)		
PCbev		100%		35%	65%	11.8		n/a	
PC		100%		36%	64%	8.8		n/a	
**Huang et al.**									6–12 >12
PT	605	100%		34.2%	65.8%	n/a		33.6	1.00 1.00
PD	204	100%		57.8%	42.2%	n/a		28.2	1.39 0.94
TOP	109	100%		75.2%	24.8	n/a		21.8	
PLD	120	100%		80.8%	19.2%	n/a		17.0	

N: patient number, TFIp: platinum therapy-free interval, HR: hazard ratios, OS: overall survival, PC: platinum combination, PD: platinum and pegylated liposomal doxorubicin, PFS: progression-free survival, PGbev: platinum, gemcitabine, and bevacizumab, PG: platinum and gemcitabine, PT: platinum and paclitaxel, PLD: pegylated liposomal doxorubicin, TOP: topotecan, TP: platinum and topotecan, PCbev: platinum-based chemotherapy with bevacizumab, PC: platinum-based chemotherapy, GC: gemcitabine with carboplatin, PDbev: pegylated liposomal doxorubicin, n/a: not available.

**Table 3 ijerph-18-06629-t003:** Multivariate Cox regression model of overall survival (OS) for 504 and 534 recurrent advanced EOC patients with TFIp between 6 and 12 months and TFIp >12 months.

Regimens	2-Year			5-Year		
	HR *	95% CI	*p*	HR*	95% CI	*p*
**TFIp 6–12 months**						
OS						
PT	1.00	(Reference)		1.00	(Reference)	
PD	1.39	0.98–1.99	0.067	1.36	1.03–1.79	0.031
TOP	1.71	1.19–2.45	0.0035	1.41	1.06–1.89	0.018
PLD	2.30	1.66–3.19	<0.0001	1.86	1.42–2.43	<0.0001
**TFIp > 12 months**						
OS						
PT	1.00	(Reference)		1.00	(Reference)	
PD	0.94	0.59–1.50	0.79	1.13	0.82–1.57	0.46
TOP	1.64	0.85–3.18	0.014	1.36	0.80–2.28	0.25
PLD	1.46	0.67–3.15	0.34	1.59	0.96–2.61	0.07

OS: overall survival, EOC: epithelial ovarian carcinoma, TFIp: platinum therapy-free interval, PT: platinum and paclitaxel, PD: platinum and liposomal doxorubicin, TOP: topotecan, PLD: liposomal doxorubicin, * hazard ratios were adjusted by age, hospital level, and residential areas.

**Table 4 ijerph-18-06629-t004:** Five-year OS of 1038 recurrent advanced EOC patients, stratified by platinum-based combinational or non-platinum single regimen with different platinum therapy-free intervals.

Regimens	Platinum (N = 809)			Non-Platinum (N = 229)		
	HR *	95% CI	*p*	HR*	95% CI	*p*
**Five-year OS**						
TFIp 6–12 months	1.00	(Reference)		1.00	(Reference)	
TFIp > 12 months	0.57	0.47–0.68	<0.001	0.56	0.38–0.82	0.003

OS: overall survival, EOC: epithelial ovarian carcinoma, TFIp: platinum therapy-free interval, platinum-based double regimens: platinum and paclitaxel or liposomal doxorubicin, non-platinum-based single regimen: topotecan or liposomal doxorubicin, HR: hazard ratio, * hazard ratio was adjusted by age, hospital level, and residential areas.

## Data Availability

The data that support the findings of this study are available on request from the corresponding author. The data are not publicly available due to privacy or ethical restrictions.

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
