# Peer review of "Chemotherapeutic Regimens and Chemotherapy-Free Intervals Influence the Survival of Patients with Recurrent Advanced Epithelial Ovarian Carcinoma: A Retrospective Population-Based Study"

_ijerph, 2021, doi:10.3390/ijerph18126629_

Round 1
Reviewer 1 Report
The pegylated lysosoma form of doxorubicin (PLD) have been using for the treatement of ovarian cancer, especially because of the reduced toxicity. The addition of doxorubicin to platinum+paclitaxel have already been shown not to improve survival (Gynecology Oncology Spirtos, N.M. 2019 and J. Clinical Oncol, Pignata, S. 2011). However, PLD has been approved by FDA for treating recurrent disease in combination with carboplatin as alternative, especially for patients that refuse alopecia (Calypso trial).
The authors contribution to the field is more relevant when considering the Chemotherapy-free interval (CFI), where they showed that patients with CFIs>12 months had better outcomes that those with CFIs of 6-12 months with liposomal doxirubicin treatments.
The tables could be improved to increase clarity and several typos were found in the manuscript that need to be fixed.
Author Response
The pegylated lysosoma form of doxorubicin (PLD) have been using for the treatement of ovarian cancer, especially because of the reduced toxicity. The addition of doxorubicin to platinum+paclitaxel have already been shown not to improve survival (Gynecology Oncology Spirtos, N.M. 2019 and J. Clinical Oncol, Pignata, S. 2011). However, PLD has been approved by FDA for treating recurrent disease in combination with carboplatin as alternative, especially for patients that refuse alopecia (Calypso trial).
Ans.: Thank you for your comments.
The authors contribution to the field is more relevant when considering the Chemotherapy-free interval (CFI), where they showed that patients with CFIs>12 months had better outcomes that those with CFIs of 6-12 months with liposomal doxirubicin treatments.
Ans.: Thank you for your comments.
The tables could be improved to increase clarity and several typos were found in the manuscript that need to be fixed.
Ans.: Thank you for your comments. We have revised the tables and corrected several typos in the manuscript. (Please see the revised tables and manuscript)
Reviewer 2 Report
Review of the manuscript ID: ijerph-1180797, entitled " Chemotherapeutic Regimens and Chemotherapy-Free Intervals Influence the Survival of Patients with Recurrent Advanced Epithelial Ovarian Carcinoma ." by Hsin-Ying Huang et al.
In this manuscript, authors discussed about the chemotherapeutic regimens for ovarian cancer patients with platinum sensitive relapse. They concluded that the PT regimen is recommended for patients with CFI of 6-12 months and the PD regimen is for patients with CFI > 12 months.
General comments
The strength of this study was the research in large number population-based cohort, and the weakness of this study was retrospective study lacking details other than regimens and survival data.
Additional data and discussion seems necessary to improve the quality of this article as below.
Specific comments
1) The authors discussed about the second line chemotherapeutic regimen by using survival period after recurrence. Recently, progression-free survival (PFS) supported by additional data, such as time to second subsequent therapy (TSST) or time until definitive deterioration of quality of life (TUDD), is the preferred endpoint rather than OS when the expected median OS is > 12 months in recurrent ovarian cancer study especially (Wilson MK et al. Ann Oncol. 2017, 28: 727-732.). Moreover, they made no mention of treatment beyond third line therapies, which has significant impact on OS. Therefore, I think that this analysis method is nappropriate for determining the second line regimen.
2) In this study the authors showed “among patients with CFIs >12 months had similar 5-year SR when treated with the respective chemotherapy regimens” in results (P8, line31). However, they described “For patients with CFIs > 12 months, the PD regimens can be the first choice” in discussion (P11, line181). I think the conclusion from the results of this study was that any regimen can be selected among patients with CFIs >12 months. The authors must explain about the reason why PD regimen is superior to non- platinum single agent.
3) In characteristics (table1), the important prognostic factors for recurrent ovarian cancer, such as initial stage, histological subtype and status of secondary debulking surgery were missing. The authors should show the detail background of patients.
Author Response
In this manuscript, authors discussed about the chemotherapeutic regimens for ovarian cancer patients with platinum sensitive relapse. They concluded that the PT regimen is recommended for patients with CFI of 6-12 months and the PD regimen is for patients with CFI > 12 months.
Ans.: Thank you for your comments.
General comments
The strength of this study was the research in large number population-based cohort, and the weakness of this study was retrospective study lacking details other than regimens and survival data.
Ans.: Thank you for your comments.
Additional data and discussion seems necessary to improve the quality of this article as below.
Specific comments
1) The authors discussed about the second line chemotherapeutic regimen by using survival period after recurrence. Recently, progression-free survival (PFS) supported by additional data, such as time to second subsequent therapy (TSST) or time until definitive deterioration of quality of life (TUDD), is the preferred endpoint rather than OS when the expected median OS is > 12 months in recurrent ovarian cancer study especially (Wilson MK et al. Ann Oncol. 2017, 28: 727-732.). Moreover, they made no mention of treatment beyond third line therapies, which has significant impact on OS. Therefore, I think that this analysis method is inappropriate for determining the second line regimen.
Ans.: Thank you for your comments.
- We have changed our terminology from survival after recurrence (SR) into overall survival (OS) to avoid misunderstanding. (Please see the revised manuscript)
- We apologize that because this was a retrospective population-based study from national database, it is not likely to get and then to analyze the treatment beyond third line therapies, TSST, or TUDD. The merit of our study was to show the real world data of the treatment and response of current different chemotherapeutic regimens in platinum sensitive recurrent ovarian cancer patients with a large number of patients. Some clinical trials on platinum sensitive recurrent ovarian cancer only published the PFS data without the OS data. So, we have added one table to emphasize the OS between the other studies and our results. (Please see Lines 222-225 and 339-341, and Pages 38-39, Table 4)
2) In this study the authors showed “among patients with CFIs >12 months had similar 5-year SR when treated with the respective chemotherapy regimens” in results (P8, line31). However, they described “For patients with CFIs > 12 months, the PD regimens can be the first choice” in discussion (P11, line181). I think the conclusion from the results of this study was that any regimen can be selected among patients with CFIs >12 months. The authors must explain about the reason why PD regimen is superior to non- platinum single agent.
Ans.: Thank you for your comments.
- We have revised our conclusions by your suggestion. (Please see Lines 48-52 and line 346-352)
- We have explained the reason why PD regimen is superior to non-platinum single agent. (Please see Lines 274-277)
3) In characteristics (table1), the important prognostic factors for recurrent ovarian cancer, such as initial stage, histological subtype and status of secondary debulking surgery were missing. The authors should show the detail background of patients.
Ans.: Thank you for your comments. We apologized that because this was a retrospective population-based study from national database, it is not likely to get and then to analyze the following items including initial stage, histological subtype and status of secondary debulking surgery. We have discussed them as our limitation. (Please see Lines 334-337)
Reviewer 3 Report
In this article the Authors aimed to evaluate factors influencing the outcomes of patients with platinum-sensitive recurrent advanced epithelial ovarian carcinoma. The manuscript is well written and has some merit. However, I think that it does not provide any significant contribution to the existing body of knowledge in the field.
Major issues
This retrospective study only confirms the results of several randomized studies published in the last two decades.
The algorithm that the Authors developed does not take into account the recommendation of the fifth Ovarian Cancer Consensus Conference (OCCC) held in Tokyo in 2015, and the dramatic evolution of the treatment for epithelial ovarian carcinoma (EOC) in recent years. As a matter of fact the Consensus suggests a tailored approach for patients with recurrent EOC, based on several factors. Furthermore, at least for high-income countries, regimens for recurrent disease must include antiangiogenic agents and PARP inhibitors, according to recent evidence.
Minor issues
Pag. 2, Line 61
The Authors state:
'When EOC recurs, the chemotherapy-free interval (CFI) … is an important factor for choosing the optimal regimen following salvage chemotherapy.'
Since the 5th Consensus it has been recognized that there are several other factors that should be considered in recurrent disease, such as tumour biology/histology, prior response, persistent toxicity, symptoms, patient‘s preferences, etc. (see Wilson et al., Annals of Oncology 2017, 28: 727–732).
Considering the linear relationship between extended PFI and platinum sensitivity, the Consensus recommends reporting PFI following primary chemotherapy as a continuous variable, rather than adopting an arbitrary definition of ‘platinum-sensitive’ or ‘platinum-resistant’ disease. (see McGee et al., Annals of Oncology, 2017, 28:702-710).
The last Gynecologic Cancer Intergroup consensus conference proposed to replace PFI with the term therapy‐free interval (TFI), dividing it into platinum‐TFI (TFIp), nonplatinum‐TFI (TFInp), and biologic agent‐TFI (TFIb) to better define different trial populations.
Pag. 2, Line 75
The Authors state:
'Therefore, extending the platinum-free interval (PFI) with a non-platinum-based regimen seems to improve survival outcomes.'
This statement is in contradiction with the following:
Pag. 9, Line 69
'it is unknown if extending the PFI with non-platinum agents can improve the response after re-introducing platinum-based chemotherapy, resulting in survival benefits in these patients.'
Current clinical evidence can neither support nor deny the benefit of extending PFI in patients with recurrent ovarian cancer (see reference # 34, Tomao et al., Cancer 2017).
Pag. 3, Line 106
The Authors state that patients with recurrent advanced EOC were administered the four major chemo- therapy regimens, as recommended by clinical guidelines in Taiwan.
Where are these guidelines published?
Line 109 and 110
The reference # 17 seems not to be appropriate
Line 138
Which were the criteria used to administer Platinum or non-Platinum based regimens, single agent or combination?
Pag. 6, Table 1
Stratifying for age 61-70 and >70 could provide more significant information.
Pag. 8, Table 2
Hazard ratios should be adjusted by other factors, such as performance status, histological subtype, etc.
Pag. 9
The Discussion is too long. Many concepts are repeated.
Pag. 11, line 169
The Authors should recognize that the retrospective studies usually have several limitations.
What are the limitations of their study?
Author Response
In this article the Authors aimed to evaluate factors influencing the outcomes of patients with platinum-sensitive recurrent advanced epithelial ovarian carcinoma. The manuscript is well written and has some merit. However, I think that it does not provide any significant contribution to the existing body of knowledge in the field.
Ans.: Thank you for your comments.
Major issues
This retrospective study only confirms the results of several randomized studies published in the last two decades.
The algorithm that the Authors developed does not take into account the recommendation of the fifth Ovarian Cancer Consensus Conference (OCCC) held in Tokyo in 2015, and the dramatic evolution of the treatment for epithelial ovarian carcinoma (EOC) in recent years. As a matter of fact the Consensus suggests a tailored approach for patients with recurrent EOC, based on several factors. Furthermore, at least for high-income countries, regimens for recurrent disease must include antiangiogenic agents and PARP inhibitors, according to recent evidence.
Ans.: Thank you for your comments. We agree that the treatment of recurrent ovarian diseases must include antiangiogenic agents and PARP inhibitors based on recent several phase III trials. The purpose of this study was to reflect the real world population data of using chemotherapy alone in the treatment of recurrent ovarian diseases. The merit of our study was to show the real world data of using chemotherapy alone in the treatment of recurrent ovarian diseases with a large number of patients. So, we have added one table to emphasize the OS between the other studies and our results. We have discussed the importance of using anti-angiogenic therapy and PARPi for the treatment of ovarian cancer. And we will further analyze the real world population data of using chemotherapy combined with antiangiogenic agents and/or PARP inhibitors in the future, when the data are mature and available. So, we have added one table to emphasize the OS between the other studies and our results. (Please see Lines 222-225 and 339-341, and Pages 38-39, Table 4)
Minor issues
Pag. 2, Line 61
The Authors state:
'When EOC recurs, the chemotherapy-free interval (CFI) … is an important factor for choosing the optimal regimen following salvage chemotherapy.'
Since the 5th Consensus it has been recognized that there are several other factors that should be considered in recurrent disease, such as tumour biology/histology, prior response, persistent toxicity, symptoms, patient‘s preferences, etc. (see Wilson et al., Annals of Oncology 2017, 28: 727–732).
Ans.: Thank you for your comments. We have discussed factors which could influence the outcome of recurrent ovarian cancer patients. (Please see Lines 334-337)
Considering the linear relationship between extended PFI and platinum sensitivity, the Consensus recommends reporting PFI following primary chemotherapy as a continuous variable, rather than adopting an arbitrary definition of ‘platinum-sensitive’ or ‘platinum-resistant’ disease. (see McGee et al., Annals of Oncology, 2017, 28:702-710).
Ans.: Thank you for your recommendation. We have revised them by your suggestion. (Please see Lines 78-89)
The last Gynecologic Cancer Intergroup consensus conference proposed to replace PFI with the term therapy‐free interval (TFI), dividing it into platinum‐TFI (TFIp), nonplatinum‐TFI (TFInp), and biologic agent‐TFI (TFIb) to better define different trial populations.
Ans.: Thank you for your recommendation. We have revised them by your suggestion. (Please see the revised manuscript)
The Authors state:
'Therefore, extending the platinum-free interval (PFI) with a non-platinum-based regimen seems to improve survival outcomes.'
This statement is in contradiction with the following:
Pag. 9, Line 69
Ans.: Thank you for your comments. We have revised them. (Please Lines 93-95)
'it is unknown if extending the PFI with non-platinum agents can improve the response after re-introducing platinum-based chemotherapy, resulting in survival benefits in these patients.'
Current clinical evidence can neither support nor deny the benefit of extending PFI in patients with recurrent ovarian cancer (see reference # 34, Tomao et al., Cancer 2017).
Ans.: Thank you for your comments. We have discussed them . (Please see Lines 260-280)
Pag. 3, Line 106
The Authors state that patients with recurrent advanced EOC were administered the four major chemo- therapy regimens, as recommended by clinical guidelines in Taiwan.
Where are these guidelines published?
Ans.: Thank you for your question. We have cited it. (Please see Lines 125 and 443-444)
Line 109 and 110
The reference # 17 seems not to be appropriate
Ans.: Thank you for your comment. We have added the appropriate reference. (Please see Line 103)
Line 138
Which were the criteria used to administer Platinum or non-Platinum based regimens, single agent or combination?
Ans.: Thank you for your questions. We apolagozed that because this is a retrospective population based manuscript. So we could not know the criteria used to administer platinum or non-platinum based regimens.
Pag. 6, Table 1
Stratifying for age 61-70 and >70 could provide more significant information.
Ans.: Thank you for your recommendation. We have revised it. (Please see the revised Table 1)
Pag. 8, Table 2
Hazard ratios should be adjusted by other factors, such as performance status, histological subtype, etc.
Ans.: Thank you for your recommendation. We apologized that because this was a retrospective population data. It is difficult to get the performance status or histological subtype from the database for further analysis. However, we have mentioned them in the limitation of this study in discussion. (Please see Lines 328-339)
Pag. 9
The Discussion is too long. Many concepts are repeated.
Ans.: Thank you for your recommendation. We have shortened the discussion. (Please see the “Discussion”)
Pag. 11, line 169
The Authors should recognize that the retrospective studies usually have several limitations.What are the limitations of their study?
Ans.: Thank you for your question. We have mentioned the limitations of our study in the discussion. (Please see Lines 328-339)
Round 2
Reviewer 2 Report
To author,
The following two issues I pointed out in the first review had not been improved sufficiently.
I recognize that it is current consensus that in the population with recurrent ovarian cancer, the effect of chemotherapy should be assessed by not survival time but time to progression, because it is difficult to distinguish whether the impact on survival is due to second-line regimen or third-line and subsequent regimen. Your real world OS data is important but it might not depended on only second line chemotherapy effect.
In addition, the results of this study show that there is no difference of 5-year OS in patients with TFIp > 12 months between platinum-based combination and non-platinum single agent. If the author believes your results, you should recommend non-platinum single agent. So, you must discuss about the reason why your results were differed from MITO-8. After that, I don’t disagree you when you finally concluded platinum doublet is recommended.
Author Response
The following two issues I pointed out in the first review had not been improved sufficiently.
I recognize that it is current consensus that in the population with recurrent ovarian cancer, the effect of chemotherapy should be assessed by not survival time but time to progression, because it is difficult to distinguish whether the impact on survival is due to second-line regimen or third-line and subsequent regimen. Your real world OS data is important but it might not depended on only second line chemotherapy effect.
In addition, the results of this study show that there is no difference of 5-year OS in patients with TFIp > 12 months between platinum-based combination and non-platinum single agent. If the author believes your results, you should recommend non-platinum single agent. So, you must discuss about the reason why your results were differed from MITO-8. After that, I don’t disagree you when you finally concluded platinum doublet is recommended.
Ans: Thank you for your comments.
- We have revised the conclusions based on our results by your suggestions. (Please see Page 2, Line 49 and Page 3, Lines 50-52; Page 20, Lines 347-352)
- We have discussed the reasons which might influence the MITO-8 and our results (Page 15, Lines 263-269 and Page 16, Lines 270-272)
Reviewer 3 Report
I would like to thank the authors for their efforts in responding to the reviewer’s comments. As consequence the manuscript was significantly improved.
Title
The article title should clearly indicate that the study is a retrospective research.
Patients and Methods
Pag. 3. Line 108
The authors state: ‘… who were administered the following four major chemotherapy regimens,as recommended by clinical guidelines in Taiwan (Taiwan Cooperative Oncology Group, year: 2011, title: Gynecological cancer therapeutic guidelines, access year: 2021, date: May 22nd.‘
These guidelines were published in July 2004, revised in August 2007, 3rd revised in 2011.
However, ‘…patients who received gynecological surgery between 1 January 2000 and 31 December 2011 were included in the study.’
Discussion
Pag. 11. Line 329
The authors state: ‘However, not all patients have BRCA mutation and PARPi would not be considered currently in this condition.’
Several studies have shown that PARPi account for a significant improvement of PFS in both recurrent and primary OC patients, independently from patients’ BRCA mutational status. See for review: Ruscito et al. Cancer Treatment Reviews, 2020;87:102040.
Pag. 11. Line 329
The authors state: ‘Low- and Middle-Income Countries (LMIC) may not have accessibility to these new agents and those that have access to these agents might still have issues with reimbursement and costs.’
However, it has recently been demonstrated that surgery may be offered in selected patients (Du Bois et al., Journal of Clinical Oncology 2020 38:15).
This option should be included in the algorithm that the Authors developed.
Conclusion
The author should clearly state that their proposed treatment of patients with recurrent ovarian cancer should only be used in countries where new agents are not available and for patients not suitable for surgery.
Author Response
I would like to thank the authors for their efforts in responding to the reviewer’s comments. As consequence the manuscript was significantly improved.
Ans: Thank you for your comment.
Title
The article title should clearly indicate that the study is a retrospective research.
Ans: Thank you for your suggestion. We have revised the title by your suggestion. (Please see Title Page, Lines 1-3)
Patients and Methods
Pag. 3. Line 108
The authors state: ‘… who were administered the following four major chemotherapy regimens,as recommended by clinical guidelines in Taiwan (Taiwan Cooperative Oncology Group, year: 2011, title: Gynecological cancer therapeutic guidelines, access year: 2021, date: May 22nd.‘
These guidelines were published in July 2004, revised in August 2007, 3rd revised in 2011.
However, ‘…patients who received gynecological surgery between 1 January 2000 and 31 December 2011 were included in the study.’
Ans: Thank you for your comments. We have revised the reference by your suggestion. Patients who were treated between January 2000 and July 2004 received the same chemotherapeutic regimens based on the consensus of gynecologic society although there was no clinical guideline at that time. The formal clinical guidelines were established since July 2004. (Please see Pages 29 and 30, reference 21)
Discussion
Pag. 11. Line 329
The authors state: ‘However, not all patients have BRCA mutation and PARPi would not be considered currently in this condition.’
Several studies have shown that PARPi account for a significant improvement of PFS in both recurrent and primary OC patients, independently from patients’ BRCA mutational status. See for review: Ruscito et al. Cancer Treatment Reviews, 2020;87:102040.
Ans: Thank you for your comments. We have revised it by your suggestion. (Please see Page 18, Lines 321-323; Page 32, reference 35)
Pag. 11. Line 329
The authors state: ‘Low- and Middle-Income Countries (LMIC) may not have accessibility to these new agents and those that have access to these agents might still have issues with reimbursement and costs.’
However, it has recently been demonstrated that surgery may be offered in selected patients (Du Bois et al., Journal of Clinical Oncology 2020 38:15).
This option should be included in the algorithm that the Authors developed.
Ans: Thank you for your recommendation. We have revised it by your suggestion. (Please see Page 35, Legend of Figure 3 and revised Figure 3)
Conclusion
The author should clearly state that their proposed treatment of patients with recurrent ovarian cancer should only be used in countries where new agents are not available and for patients not suitable for surgery.
Ans: Thank you for your recommendation. We have revised our conclusions by you and the other reviewer’s opinions. (Please see Page 2, Line 49 and Page 3, Lines 50-52; Page 20, Lines 347-352; Page 35, Legend of Figure 3 and revised Figure 3)